# OpenReview forum: "Decision-Making Behavior Evaluation Framework for LLMs under Uncertain Context"
_NeurIPS.cc/2024/Conference — NeurIPS 2024 poster_

### Official Review · Reviewer_hBFU · 2024-07-09

**Soundness:** 4
**Presentation:** 4
**Contribution:** 3
**Rating:** 6
**Confidence:** 4

**Summary:**

This paper proposes a comprehensive framework to evaluate LLM's decision-making behavior under uncertain contexts. By leveraging computational models from behavioral economics, as well as the experiment paradigms, the authors investigated the risk preferences, probability weighting, and loss aversion of LLMs from the parameters in the models. Finally, by injecting demographic information into LLMs, the authors found significant differences in those parameters measured from behaviors, suggesting potential biases from demographic information (e.g., age, education, or gender) on decision-making tendencies.

**Strengths:**

- This paper advances the evaluation framework of LLM's decision-making behavior under uncertainty, borrowing the models from behavioral economic theories to quantitatively model the behavior of LLMs. By leveraging risk preference, probability weighting and loss aversion, the authors have a deeper investigation on risky decision-making behaviors.

- This paper also investigates demographic influences on the LLMs' behaviors, suggesting potential biases in generating behaviors under different demographic settings.

**Weaknesses:**

- __Lack of human data__. As the authors have mentioned in the limitation section, it would be fun to investigate the LLM vs. Human behavior at the parameter level of computational models. These parameter differences might explain why humans and LLMs are different or similar on a deeper level.

- __Lack of model selections__. The authors ONLY use one model (TCN model) to evaluate the LLM's behaviors. In computational cognitive sciences and psychology (Wilson, Collins, 2019), it is common to propose multiple cognitive models to test which model might best explain the behaviors of humans (and of machines). The authors could strengthen the model evidence by proposing multiple competing models, comparing models, showing goodness-of-fitting, and even possibly simulation and parameter recovery to show the robustness of the currently selected model. Perhaps there might be better models that can explain the behaviors better and thus, the attribution of behaviors may change.

- __More investigations on LLMs are recommended__. The authors list one of the contributions as investigating multiple LLMs in this framework. The LLM selections are the most known commercial LLMs (like GPT-4, Claude, and Gemini). These model comparisons somehow make sense but two of them are not open-sourced and the training data are not transparent. If the authors could compare some LLMs with a more transparent basis to provide insights into LLMs, this would strengthen the contribution of the paper. For example, the authors could investigate the relationship between model size and the parameters measured from the behaviors. Choosing models within the same family, and trained on similar datasets would make comparisons more informative. Looking at prompt engineering, the way of instruction-tuning or model structure (e.g., encoder-decoder model or decoder-only model) can also make sense while making this framework provide more information.

**Questions:**

- One interesting finding from the paper is that human studies may not find significant differences in risk preferences in sexually minority groups but LLMs do respond differently under these settings. There might be two but completely different reasons: first, the authors did not get the full knowledge of the literature in that field and the paper listed there may be only one study that reports non-significant research. The other reason could be intriguing: the studies about one specific topic are limited but humans are generally exhibiting such effects which are overlooked in empirical study. However, LLMs are more likely to learn such effects from massive datasets and thus exhibit effects. So there might be a reverse approach that understanding human behaviors (especially in fields that are overlooked) from LLM's behavior. The authors are not required to formally give an answer to this question, but I find it interesting in public discussion about such potentials.

**Limitations:**

As indicated above, the major limitations of the paper are not providing a deeper insight into some human cognitive or LLM mechanistic findings, though this framework is valuable and more comprehensive than previous work.

---

> ### Author Rebuttal · Authors · 2024-08-07
>
> We appreciate your thoughtful review and the detailed feedback provided. Below, we address each of the concerns and questions raised in the weaknesses section.
>
> 1. Lack of Human Data:
>
> We agree that including more human data would significantly enhance our study. The bottom row in Table 6 shows human parameter data retrieved from one of the original studies that adapted the same evaluation model on human subjects. We acknowledge that we did not emphasize human data in this work due to the complexity and variability inherent in human studies. Human data are influenced by numerous factors, including cultural, socioeconomic, and environmental contexts. Additionally, existing human studies often do not represent the entirety of the human population. For instance, some studies might use college students as samples, while others use rural households, each occurring in different countries and involving people with diverse backgrounds.
> Integrating such diverse datasets requires the involvement of sociology and economic theories capable of accounting for these variations and ensuring that the resulting analysis is both representative and meaningful. Moreover, ethical considerations are crucial, especially when studying minority groups.
>
> We are currently collaborating with behavioral economics experts on ongoing research to compare LLM and human data for an apple-to-apple comparison. We hope to complete this work in the near future and believe it will add significant value to our study.
>
> 2. Lack of Model Selection:
>
> For model selection, we conducted a comprehensive literature review. In the domain of decision-making behavior under risky contexts, foundational theories are Expected Utility Theory (EUT) and Prospect Theory (PT). Various models based on these frameworks have been developed for human subjects. While previous studies have used Prospect Theory models, we discussed their limitations in our paper, particularly the issue of pre-assumptions. The TCN model combines elements of both EUT and PT without relying on these pre-assumptions, making it particularly suitable for testing LLMs. This was the primary reason for our adoption of this model.
>
> Although our current paper is limited by space constraints, we see great value in your suggestions and will include related results of more models in the final manuscript if accepted and consider them in our future works. We are actively developing new models that may better suit the evaluation of LLM decision-making processes. If our paper is accepted, we will also include a section on future work to discuss these potential directions, integrating more evaluation models and data to enhance the robustness and utility of our framework.
>
> 3. Lack of LLM Variety:
>
> We truly consider your suggestions regarding this to be valuable. The ideas of investigating the relationship between model size and behavior parameters, comparing models within the same family and trained on similar datasets, and examining factors like prompt engineering and model structures, are all meaningful and important research questions to be explored.  While the primary goal of this study is to propose a framework for evaluating decision-making behavior in LLMs, rather than to exhaustively test all models, we are excited to incorporate these directions in future work to enhance the robustness and informativeness of our framework.
>
> 4. Response to the Question:
>
> Thank you for your interest in this finding regarding the differences in risk preferences between LLMs and human studies. This approach that tries to understand human behaviors from LLM behaviors could indeed be intriguing and valuable for public discussion. We touched on this topic slightly in our discussions and did our best to review relevant literature.
>
> Studying minority groups is always challenging in social sciences, given the context and biases such as sample size limitations and societal prejudices. If it is evident that LLMs have the ability to serve as a supplemental tool, they could help overcome some of these challenges by identifying patterns and preferences that are difficult to detect in traditional studies. Our work aims to serve as an evaluation framework for both researchers and users to at least reveal some information and subsequently infer human characteristics. Leveraging the vast datasets from LLMs, we hope the use can potentially gain a deeper understanding of human behaviors, particularly in under-researched areas, and enrich the field of social sciences.
>
> Thank you again for your kind review. We hope our work can be accepted and become a useful evaluation tool for all end-users and researchers.

---

> > ### Comment · Reviewer_hBFU · 2024-08-11
> >
> > Thanks to the authors for the rebuttal. I think the comments well addressed my concerns and positively engaged in discussions. My original evaluation is to have this paper present in NeurIPS and thus I will maintain my current evaluation, given the constraint of the overall scope. However, I do think this paper has a lot of interesting points and may raise more valuable research in the future.

---

> > > ### Author Response · Authors · 2024-08-11
> > >
> > > We appreciate your positive evaluation and are glad our rebuttal addressed your concerns. Thank you for your thoughtful feedback and support. We look forward to contributing to further research in this area.

---

### Official Review · Reviewer_YNJ7 · 2024-07-11

**Soundness:** 3
**Presentation:** 3
**Contribution:** 3
**Rating:** 4
**Confidence:** 3

**Summary:**

This paper presents a novel framework for evaluating the decision-making behavior of large language models (LLMs) under uncertain contexts, grounded in behavioral economic theories. The authors conducted experiments to estimate risk preference, probability weighting, and loss aversion for three commercial LLMs: ChatGPT-4.0-Turbo, Claude-3-Opus, and Gemini-1.0-pro. The study further explores the impact of embedding socio-demographic features on the decision-making process of these models, revealing significant variations and potential biases. The paper concludes with a call for the development of standards and guidelines to ensure ethical and fair decision-making by LLMs.

**Strengths:**

1. Introduces a comprehensive framework for evaluating LLMs' decision-making behavior, which is the first application of behavioral economics to LLMs without preset behavioral tendencies.
2. Provides empirical evidence of LLMs' tendencies towards risk aversion and loss aversion, with a nuanced approach to probability weighting, offering insights into their alignment with or divergence from human behavior.
3. Conducts further experiments with socio-demographic feature embeddings, uncovering disparities in decision-making across various demographic characteristics, which is crucial for understanding potential biases in LLMs.

**Weaknesses:**

1. The paper does not provide a detailed analysis of the potential causes behind the observed variations in LLM behavior when demographic features are introduced, which could be crucial for understanding and mitigating biases.
2. The study's focus on three commercial LLMs may limit the applicability of the findings to some open source LLMs like LLaMA.
3. The authors should discuss the references like chain-of-thought (CoT) prompting and reasoning in LLMs, such as "Chain-of-Thought Prompting Elicits Reasoning in Large Language Models" (https://arxiv.org/abs/2201.11903), "Automatic Chain of Thought Prompting in Large Language Models" (https://arxiv.org/abs/2210.03493), and "Keypoint-based Progressive Chain-of-Thought Distillation for LLMs" (https://arxiv.org/abs/2405.16064). These works could provide additional context and depth to the understanding of LLM decision-making processes.’

**Questions:**

Please see the Weakness.

**Limitations:**

Yes

---

> ### Author Rebuttal · Authors · 2024-08-07
>
> We appreciate your thorough review and insightful comments. Below, we address each of the questions raised in the weaknesses section.
>
> 1. Regarding the Analysis of Potential Causes for Observed Variations:
>
> As mentioned in our discussion section (lines 308-313), we have identified several potential reasons for the observed discrepancies between human and LLM behaviors from ethical and sociological perspectives:
>
> (1) Flaws in Human Studies: Studies on minority groups often face privacy challenges in obtaining representative samples, leading to potential biases and flawed understandings.
>
> (2) LLM Misunderstanding: LLMs might possess biases from data and training, which could affect their decision-making behavior.
>
> We focused our discussion from the data and training perspective because we believed whichever algorithms the LLMs use to update their parameter would’ve inevitably introduced bias, as long as the data wasn’t “perfectly just”. We suggested that if LLMs could reflect real-world trends more accurately with broader, cleaner data and careful training, they might be valuable for social studies, but further research is needed to address human-LLM discrepancies.
> While exploring the algorithmic details of model architecture or training datasets is beyond the current paper's scope, we recognize the importance of algorithm-level analysis and will add to our discussion section a recommendation for future research to study the underlying model architectures and training data. Recent algorithm-level studies, such as Meade et al. (2021), Bender et al. (2020), Zhang et al. (2024), Liu et al. (2024), and Guo et al. (2022), as well as the papers related to Chain-of-Thought (CoT) prompting (Wei (2022), Zhang (2022), Feng (2024)), provide foundational insights into addressing biases in LLMs. These works suggest techniques like adversarial debiasing, data augmentation, and fairness constraints, which could be explored in future studies to mitigate biases in LLMs.
>
> Our aim is to serve as an evaluation tool of the behavioral aspects of LLMs in specific contexts. However, we hope that our work provides a robust foundation for such evaluations, facilitating a deeper understanding and mitigation of biases in LLM decision-making behaviors. Understanding the exact causes of LLM behavior remains a complex question; it also cannot be done without evaluating a multitude of models with a foundational framework like ours.
>
> 2. Regarding the Inclusion of Commercial LLMs:
> Thank you for your suggestion to include more models, especially the open-source models; we also believe this is important work for this research area. We also have carefully planned that for the future work, we will include more models, and testing various dimensions such as architecture differences, model size differences, and dataset differences.
>
> We specifically chose to evaluate these three commercial LLMs due to their uniqueness and closed-source nature. This choice underscores the need for a reliable and stable evaluation framework to ensure these models behave as intended by their developers. Our framework serves as the first step toward this goal.
>
> Nevertheless, comparing LLMs with more transparent/open-sourced bases would indeed strengthen the contribution of our paper.  For example, investigating the relationship between model size and behavior parameters, or examining prompt engineering and instruction-tuning techniques, could provide more detailed insights. While these extensions are beyond the scope of our current study, they represent valuable directions for future research.
>
> 3. Regarding References to Chain-of-Thought Prompting and Reasoning:
>
> Thank you for the suggestions. We will incorporate discussions of the related works on CoT prompting and reasoning in our revised manuscript if accepted. These references could indeed provide additional context and depth to our understanding of LLM decision-making processes and would make our work more comprehensive.
>
>
> References:
>
> •	Bender, E. M., Gebru, T., McMillan-Major, A., & Shmitchell, S. (2020). On the Dangers of Stochastic Parrots: Can Language Models Be Too Big? Proceedings of the 2021 ACM Conference on Fairness, Accountability, and Transparency (FAccT '21).
>
> •	Guo, Y., Yang, Y., & Abbasi, A. (2022). Auto-Debias: Debiasing Masked Language Models with Automated Biased Prompts. Proceedings of the 60th Annual Meeting of the Association for Computational Linguistics (Volume 1: Long Papers).
>
> •	Meade, N., Poole-Dayan, E., & Reddy, S. (2021). An empirical survey of the effectiveness of debiasing techniques for pre-trained language models. arXiv preprint arXiv:2110.08527.
>
> •	Liu, T., et al. (2024). LIDAO: Towards Limited Interventions for Debiasing (Large) Language Models. arXiv preprint arXiv:2406.00548.
>
> •	Zhang, Y. F., et al. (2024). Debiasing large visual language models. arXiv preprint arXiv:2403.05262.
>
> •	Wei, Jason, et al. (2022). "Chain-of-thought prompting elicits reasoning in large language models." Advances in neural information processing systems 35: 24824-24837.
>
> •	Zhang, Zhuosheng, et al. (2022). "Automatic chain of thought prompting in large language models." arXiv preprint arXiv:2210.03493.
>
> •	Feng, Kaituo, et al. (2024). "Keypoint-based Progressive Chain-of-Thought Distillation for LLMs." arXiv preprint arXiv:2405.16064 .

---

### Official Review · Reviewer_1hfj · 2024-07-12

**Soundness:** 3
**Presentation:** 4
**Contribution:** 2
**Rating:** 6
**Confidence:** 4

**Summary:**

In this paper, the paper the authors model the decision-making behavior of certain open-source LLMs on lottery selection tasks. They further extend this work, by repeating similar analysis after priming the models with demographic information. Results indicate that the different LLMs differ in their risk profiles as inferred by the parameters of the evaluative model. Some prominent differences also emerge on demographic-primed LLMs. The overarching theme that comes through is the need for caution when using LLMs in practice, especially as analogue to human decision-making.

**Conclusion:** Overall, while the paper considers an interesting problem, the results don’t appear to yield any general insights, and the methodology seems to make assumptions that may be difficult to justify. The paper can still be an alright addition to the conference since it’s timely and touches on an important problem.

**Strengths:**

1. It’s clearly written, provides ample context and the ideas are developed in a streamlined fashion.
2. The problem considered is pertinent and the approach seems to be well motivated by a long history of modeling human behavior.
3. The analysis makes sense — studying the distribution of pertinent parameters conditioned on different agents, and demographics to derive insights about the underlying behavior.

**Weaknesses:**

1. It’s unclear whether the TCN model is actually apt in this setting. For example, it seems to assume deterministic values of parameters like sigma, alpha and lambda, and then tries to infer them based on the LLMs decisions. However, what if these parameters are themselves random variables? Does it make sense to assume the LLM has a fixed risk profile across trials? As an analogy, assuming the coin has a fixed bias p, and estimating it via repeated tossing experiments when p is sampled from a uniform distribution each trial.

2. The utility function was a little unclear to me (line 135). Why does it not cover the entire (x,y) domain? w(p) simplifies to 1/p for alpha = 1. Should it be p? As a tangentially related point, consider numbering the equations.

3. In Step 4 (section 4), can we be certain the estimation algorithm actually converges? This relates to 1, so that the convergence might not be guaranteed if the latent parameters are not fixed.

4. While the qualitative results are sensible — due diligence is needed before LLMs can be entirely trustworthy, the quantitative results don’t impart any deep or actionable insights. Observations like a given version of a pre-trained model tends to show risk aversion doesn’t quite generalize.

**Questions:**

NA

---

> ### Author Rebuttal · Authors · 2024-08-07
>
> We sincerely appreciate your careful review and valuable feedback. Your comments are very insightful and have provided us with the opportunity to clarify our work. Before addressing the specific questions, we would like to emphasize that we recognize that LLMs are machines, not humans. However, by considering the set of queries of LLMs as analogous to a population of humans, we can begin to evaluate their behavior by adapting well-established models tested by human experiments. This approach represents the initial phase of our studies, laying the groundwork and finally moving towards the development of models specifically for LLMs.
>
> 1. Regarding the Aptness of the TCN Model:
>
> The TCN model referenced in the paper was originally designed to evaluate human decision-making behavior, assuming each individual has fixed parameters (sigma, alpha, lambda). In human studies, these parameters are considered deterministic for each individual; while across a population, they are treated as random variables due to inter-individual variability. Our work treats LLMs as analogous to human populations, where each LLM exhibits overall tendencies.
> When applied to LLMs, each query to a specific LLM can be viewed as one interaction with a population of subjects. These subjects in the population share one distribution of the parameters. For each interaction, we assume fixed parameters specific to that interaction. This assumption applies to each particular interaction rather than the entire set of interactions or “population”. Therefore, while these parameters are fixed for a particular interaction, they may vary across different interactions, akin to different individuals in a population study.
>
> By conducting multiple rounds of experiments (e.g., 300 times), we aim to capture the population’s overall tendencies. The mean and standard deviation of these parameters across interactions provide an estimation of the general tendencies and variability of the LLMs’ behaviors.
>
> Since the study of LLMs' decision-making behavior is still in its infancy, there is a lack of literature addressing the granularity of the studies and whether their behavior should be considered as having fixed biases or as random variables, which haven’t been decided by some recent studies using simpler models either, as discussed in lines 105-112 of our paper. We also recognize the need for more comprehensive modeling, potentially involving hierarchical models. We have submitted the raw data and plan to release it, enabling other researchers to further explore the possibility of other distributions and refine models to better understand LLM behavior.
>
> 2. Regarding the Utility Function:
> (1)	Domain of (x,y)
> This utility function actually have covered the all possible cases of (x,y): our utility function determines which value should be x or y based on their signs and magnitudes, ensuring that all combinations of positive and negative values are appropriately handled. The detailed logic and examples are as follows:
> Denote two outputs O1 and O2, abs(O1) > abs(O2).
> if O1O2>0: x = O1, y=O2
> if O1O2<0: x = min(O1, O2), y = max(O1, O2).
> Our utility function is adapted from the framework in the TCN work which is extensively validated with human subjects. This adaption ensures our methodology is grounded in rigorously tested principles of behavioral economics.
> (2) Typo in the Utility Function: You are correct; upon cross-checking our mathematical derivation and code script (available in the supplementary materials), we found a typo in the equation between lines 135 and 136, where we missed a “-” in w(p). This typo led to confusion regarding the utility function. We apologize for this error and appreciate you pointing it out. We will correct the associated mathematical work throughout. Please note that our code and the results presented in the study are based on the correct form of the utility function, so the typo in the manuscript did not affect our computational results.
> (3) Numbering Equations: Thank you for suggesting numbering the equations. We will incorporate this into our revised manuscript to enhance clarity.
>
> 3. Regarding the Convergence of the Estimation Algorithm in Step 4:
>
> Each LLM interaction is treated as an independent trial with fixed parameters, as discussed in our response to the first question. In this setup, the parameters are treated as constants within one particular interaction but vary across different interactions, similar to how characteristics vary among individuals in human population studies.
>
> Given this setup, the convergence of the estimation algorithm in Step 4 is facilitated by systematically narrowing the intervals for each parameter through the iterative solution of groups of inequalities. We have three groups of inequalities and three parameters; therefore, they allow us to obtain the interval for the three parameters.
>
> Iteratively solving for the intervals ensuring that the algorithm converges by gradually refining the estimates until the intervals are sufficiently narrow so that our parameter estimates are robust and reliable across different experimental iterations.
>
> 4. Regarding the Depth of Quantitative and Qualitative Insights:
>
> In our work, the quantitative estimation of parameters, followed by regressions to determine the correlations between embedded demographic features and decision-making behavior, provides the foundation for claiming qualitative observations of a given version of a model. The observations are specific to the models, but the method to draw qualitative observation from the quantitative parameters are generalizable.
>
> Nevertheless, we acknowledge that finer insights from quantitative results will be beneficial to obtain more detailed qualitative observations. We plan to enhance our analysis granularity in our future work.

---

> > ### Comment · Reviewer_1hfj · 2024-08-12
> > **Update after authors' rebuttal**
> >
> > I appreciate the authors taking the time to offer clarification and answer a couple of of the questions I had. I agree that even though the authors only consider a few popular models, their methodology generalizes irrespective of whether the specific findings around a given LLM may or may not. I continue to believe that this is a nice interdisciplinary paper, and I am maintaining my acceptance decision along with the score.

---

> ### Author Response · Authors · 2024-08-13
>
> Dear Reviewer,
>
> Thank you for your positive feedback and for recognizing the generalizability of our methodology. We appreciate your support and are glad that our clarifications were helpful. Thank you for maintaining your acceptance decision!

---

### Official Review · Reviewer_gKiM · 2024-07-13

**Soundness:** 2
**Presentation:** 2
**Contribution:** 2
**Rating:** 4
**Confidence:** 3

**Summary:**

This paper proposes a framework to evaluation the decision making behavior of LLMs. It focuses on ChatGPT4Turbo, Claude3Opus and Gemini1pro. The results show that LLMs exhibit patterns similar to humans. Impacts of socio-demographic features are also analyzed and the results show that different LLMs can vary from each other.

**Strengths:**

- The problem in consideration is important. The findings appear interesting.

**Weaknesses:**

- While the problem is interesting, the technical contribution of the paper seems limited. The measures chosen are also not fully justified. It would be important to highlight the novelty of the results.
- Following the above comment, the examination does not appear comprehensive. The reviewer worries that the questions may only cover a small set of problems. As a result, the conclusions drawn might contain significant randomness.

**Questions:**

- The problem in consideration is important. The findings appear interesting.
- While the problem is interesting, the technical contribution of the paper seems limited. The measures chosen are also not fully justified. It would be important to highlight the novelty of the results.
- Following the above comment, the examination does not appear comprehensive. The reviewer worries that the questions may only cover a small set of problems. As a result, the conclusions drawn might contain significant randomness.

**Limitations:**

Please see the above.

---

> ### Author Rebuttal · Authors · 2024-08-07
>
> We appreciate the time and effort you have taken to review our paper and provide constructive feedback. Below, we address each of your comments and concerns:
>
> Clarification of Summary:
> Our paper presents a framework for evaluating the decision-making behavior of LLMs under uncertainty grounded in behavioral economics theories. The three models we evaluated—ChatGPT4Turbo, Claude3Opus, and Gemini1pro—serve as examples of the framework's application on commercial LLMs. To clarify, our results do not simply show that LLMs exhibit human-like patterns; they reveal significant variations across LLMs and from human in the three-dimensional parameters we evaluated. Furthermore, our framework uncovers ethical implications and biases in LLMs, such as increased risk aversion for attributes of sexual minority groups or physical disabilities in Claude3Opus.
>
> Response to Weaknesses and Questions:
> Our work provides a novel and comprehensive approach to evaluating LLMs' decision-making behavior. We also believe the measures we chose are justified, as they align with current practices in evaluating human decision-making behaviors. Highlighting the novelty of our results, our framework uncovers biases and ethical implications that were previously unexplored in LLM evaluations. Specially:
>
> (1)	About Technical Contribution:
> We consider our technical contributions to be multifaceted:
>
> 1.	This is the first work to propose a framework for evaluating LLM decision-making behavior under economic uncertainty without pre-required assumptions. This framework provides a standardized valid method for future research.
>
> 2.	Our technical contribution includes a comprehensive analysis of existing models' mathematical pre-assumptions, ensuring our chosen model is the most appropriate for creating this baseline framework.
>
> 3.	We conduct an evaluation of three state-of-the-art commercial LLM models: ChatGPT4Turbo, Claude3Opus, and Gemini1pro. This evaluation demonstrates the versatility and applicability of our framework across different leading models. We also analyze how socio-demographic characteristics influence LLM decision-making compared to humans, revealing significant insights into biases and ethical implications.
>
> To summarize, inventing a new model for LLMs is premature without first establishing a reliable baseline framework. While developing new models for LLMs ongoing and we also involve in that, without a baseline like in this paper, these new models would lack foundation and comparability.
>
> (2)	About Measurement Justification:
> For measurement selection, as in the related work section, especially line 102-116, we conducted a comprehensive literature review before determining using TCN model. In the domain of decision-making behavior under risky contexts, foundational theories are Expected Utility Theory (EUT) and Prospect Theory (PT). Various models based on these frameworks have been developed for human subjects. While previous studies have used PT models, we discussed their limitations in our paper, particularly the issue of pre-assumptions. The TCN model combines elements of both EUT and PT without relying on these pre-assumptions, making it particularly suitable for testing LLMs. This was the primary reason for our adoption of this model.
>
> (3)	About Scope of Examination:
> The study of decision-making behavior is a substantial and significant area in both behavioral economics and psychology. It encompasses a wide range of applications, from financial decisions to healthcare choices, and is crucial for understanding human behavior in uncertain contexts. For instance, pioneer works by Kahneman and Tversky (1979) and Thaler (2015) illustrate the vast importance and impact of decision-making studies on economics and psychology. These studies have provided foundational insights into how individuals evaluate risk and uncertainty, influencing a multitude of fields and practices (Kahneman & Tversky, 1979; Thaler, 2015; Camerer, 2003; Camerer & Hogarth, 1999; Rabin, 2000; Bavel et al., 2020).
>
> As LLMs are increasingly involved in decision-making processes, we believe it is time to propose a framework for evaluating them, which is of great importance. Given these examples, we hope it is clear that the scope of decision-making behavior is far from narrow and holds substantial relevance in various critical fields. The settings in this paper are well-established and widely utilized in behavioral studies, linking closely to many practical applications, and covering an exhaustive range of scenarios is beyond the scope of establishing the framework.
> Further, to address concerns about statistical randomness, we conducted each experiment for every LLM under every setting 300 times. This approach ensures a large enough sample size to mitigate the effects of randomness and variability. In behavioral economics and psychology, similar sample sizes are often used to draw robust and reliable conclusions.
>
> Final Remarks:
> We appreciate the opportunity to clarify our work and address your concerns. We hope this response provides a clearer understanding of our contributions and the scope of our study. We believe our work represents a significant step forward in the evaluation of LLMs and their decision-making behaviors.
>
> References:
>
> 1.	Kahneman, D., & Tversky, A. (1979). Prospect Theory: An Analysis of Decision under Risk.
>
> 2.	Thaler, R. H. (2015). Misbehaving: The Making of Behavioral Economics.
>
> 3.	Camerer, C. F. (2003). Behavioral Game Theory: Experiments in Strategic Interaction.
>
> 4.	Camerer, C. F., & Hogarth, R. M. (1999). The Effects of Financial Incentives in Experiments: A Review and Capital-Labor-Production Framework.
>
> 5.	Rabin, M. (2000). Risk Aversion and Expected-Utility Theory: A Calibration Theorem.
>
> 6.	Bavel, J. J. V., Baicker, K., Boggio, P. S., et al. (2020). Using social and behavioural science to support COVID-19 pandemic response. Nature Human Behaviour, 4(5), 460-471.

---

### Comment · Area_Chair_NsHb · 2024-08-08
**Reviewer - Author Discussion**

Thanks everyone for their hard work on the papers, reviews, and rebuttals. We now have a comprehensive rebuttal from the authors which responds both overall and to each review.

I'd please ask the reviewers to please post a comment acknowledging that they have read the response and ask any followup questions (if any).

This period is to be a discussion between authors and reviewers (Aug 7 - Aug 13) so please do engage now, early in the window, so there is time for a back and forth.

Thanks!

---

### Decision · Program_Chairs · 2024-09-25

**Decision:**

Accept (poster)

**Comment:**

After a review/rebuttal/discussion phase that included most of the reviewers we are recommending this paper be accepted for NeurIPS.

All reviewers agreed that the problem in the paper (evaluating decision behavior of LLMs) is an important and interesting one for AI broadly. They also agreed that the paper is clearly written, novel, and is well placed in the literature.

There were some concerns about the comprehensiveness and completeness of the proposed framework, specifically, that the chosen set of measures are not fully justified/comprehensive. We urge the authors to addresses these comments in the final by arguing more directly about both of these properties.

However, even given the above, the paper does a solid job of articulating and testing the proposed evaluation framework. So while there is always space to be more comprehensive and more general, the steps taken in the paper are a significant first step. We hope the authors will address the reviewers comments on shortcomings in the final version.